# On Improving Adaptive Problem Decomposition Using Differential Evolution for Large-Scale Optimization Problems

Aleksei Vakhnin [ID], Evgenii Sopov [ID] and Eugene Semenkin *[ID]

Department of System Analysis and Operations Research, Reshetnev Siberian State University of Science and Technology, Krasnoyarsk 660037, Russia
* Correspondence: eugenesemenkin@yandex.ru

**Abstract:** Modern computational mathematics and informatics for Digital Environments deal with the high dimensionality when designing and optimizing models for various real-world phenomena. Large-scale global black-box optimization (LSGO) is still a hard problem for search metaheuristics, including bio-inspired algorithms. Such optimization problems are usually extremely multi-modal, and require significant computing resources for discovering and converging to the global optimum. The majority of state-of-the-art LSGO algorithms are based on problem decomposition with the cooperative co-evolution (CC) approach, which divides the search space into a set of lower dimensional subspaces (or subcomponents), which are expected to be easier to explore independently by an optimization algorithm. The question of the choice of the decomposition method remains open, and an adaptive decomposition looks more promising. As we can see from the most recent LSGO competitions, winner-approaches are focused on modifying advanced DE algorithms through integrating them with local search techniques. In this study, an approach that combines multiple ideas from state-of-the-art algorithms and implements Coordination of Self-adaptive Cooperative Co-evolution algorithms with Local Search (COSACC-LS1) is proposed. The self-adaptation method tunes both the structure of the complete approach and the parameters of each algorithm in the cooperation. The performance of COSACC-LS1 has been investigated using the CEC LSGO 2013 benchmark and the experimental results has been compared with leading LSGO approaches. The main contribution of the study is a new self-adaptive approach that is preferable for solving hard real-world problems because it is not overfitted with the LSGO benchmark due to self-adaptation during the search process instead of a manual benchmark-specific fine-tuning.

**Keywords:** problem decomposition; large-scale global optimization; self-adaptive differential evolution; memetic algorithm; cooperative co-evolution.

## 1. Introduction

Modern numerical continuous global optimization problems deal with high dimensionality and the number of decision variables is still increasing because of the need to take into account more internal and external factors when designing and analyzing complex systems. This is also facilitated by the development of high-performance hardware and algorithms. "Black-box" large-scale global optimization (LSGO) is one of the most important and hardest types of optimization problems. The search space of LSGO problems exponentially grows and many state-of-the-art optimization algorithms, including evolutionary algorithms, lose their efficiency. However, the issue cannot be solved by straightforward increasing the number of objective function evaluations.

Many researchers note that the definition of a LSGO problem depends on the nature of the problem and changes over time and with the development of optimization approaches. For example, the global optimization of Morse clusters is known as a hard real-world

optimization problem. The best-found solutions for Morse clusters are collected in the Cambridge Energy Landscape Database [1]. At the moment, the database contains the highest value equal to 147 atoms, which corresponds to 441 continuous decision variables only. The most popular LSGO benchmark was proposed within the IEEE Congress on Evolutionary Computation and is used for the estimation and comparison of new LSGO approaches. The benchmark contains 1000-dimensional LSGO problems. There exist solutions for real-world problems with many thousands of decision variables.

The general LSGO optimization problem is defined as (1):

$$f(x_1, x_2, \ldots, x_n) \to \min_{x \in R^n}, f : R^n \to R^1, \tag{1}$$

here $f$ is an objective function, $x_i$ are box-constrained decision variables. We do not impose any restrictions on the type of the objective function, such as linearity, continuity, convexity, and the need to be defined at all points requested by the search algorithm. In the general case, the objective function is defined algorithmically, there is no information about the properties of its landscape, thus the objective function is a "black-box" model.

As previously mentioned, the performance of many black-box global optimization algorithms cannot be improved by only increasing the budget of function evaluations when solving LSGO problems. One of challenges for researchers in the LSGO field is the development of new approaches, which can deal with the high dimensionality. Various LSGO algorithms that use fundamentally different ideas and demonstrate different performances for different classes of LSGO problems have been proposed. When solving a specific LSGO problem, a researcher must choose an appropriate LSGO algorithm and fine-tune its parameters. Moreover, the algorithm can require different settings at different states of the optimization process (for example, at exploration and exploitation stages). Thus, the development of self-adaptive approaches for solving hard LSGO problem is an actual research task.

In this study, an adaptive hybrid approach that combines three general conceptions, such as problem decomposition using cooperative co-evolution (CC), global search based on differential evolution (DE), and local search is proposed. This approach demonstrates performance comparable with LSGO competition winners and outperforms most of them. At the same time, it demonstrates the same high efficiency for different classes of LSGO problems, which makes the proposed approach preferable for "black-box" LSGO problems when it is not possible to prove the choice of an appropriate search algorithm.

The rest of the paper is organized as follows. Section 2 presents the related works for reviewing state-of-the-art in the field of LSGO and motivates designing a hybrid approach. Sections 3 and 4 describe the proposed approach, experimental setups, some general top-level settings, and implementation. In Section 5, the experimental results, analysis, and discussion of the algorithm dynamics and convergence, and the comparison of the results with state-of-the-art and competition-winner approaches are presented. In conclusion, the proposed methods and the obtained results are summarized and some further ideas are suggested.

## 2. Related Work

The complexity of real-world optimization problems has grown in recent years and is still growing. The class of global "black-box" optimization problems for which the high dimensionally causes the loss in the performance of a search algorithm is known as Large Scale Global Optimization or LSGO. Well-known experts in the field of LSGO, Mohammad Nabi Omidvar and Xiaodong Li, note that the term "large scale" is not definitively determined because the dimensionality of problems in LSGO grows over time, and it can also be different in different application areas. Many modern metaheuristics, including EAs, consider LSGO problems with 1000 real variables.

One of the first discussions on LSGO have been proposed within the special session of the IEEE CEC conference in 2008 [2,3]. Since 2008, the LSGO scientific community has

proposed the LSGO benchmark for evaluating and comparing LSGO algorithms. The first benchmark in 2008 had only 7 test problems, including 2 unimodal and 5 multi-modal optimization problems [2]. The CEC LSGO 2010 benchmark was extended with test problems grouped by the separability property and contained 20 optimization problems, including 3 fully separable, 15 partially non-separable, and 2 fully non-separable functions [4]. Finally, within the IEEE CEC 2013 special session and completion, a new benchmark has been proposed, and it is still used today and is known as a hard benchmark set for many state-of-the-art LSGO techniques [5]. The CEC LSGO 2013 benchmark contains 3 fully separable, 8 partially additive non-separable functions, 3 functions with overlapping components, and 1 fully non-separable function. In 2018, a new online Toolkit for Automatic Comparison of Optimizers (TACO) has been proposed for a fair independent comparison of LSGO algorithms [6]. In 2021, the TACO database includes the results of 25 leading and competition-winner LSGO algorithms.

Some extensive studies with surveys of the current state of LSGO and systematizations of LSGO techniques have been proposed in [7,8]. In [9], LSGO is highlighted as one of the urgent domains of bio-inspired computation. The recent work on the LSGO review proposes a large summary of the state of affairs and accumulated experience [10,11]. Within the proposed systematizations, the following main approaches are developing:

- Random (static or dynamic) problem decomposition using cooperative co-evolution,
- Learning-based decomposition using cooperative co-evolution,
- Modifications of the standard evolutionary algorithms without problem decomposition, including hybrid memetic approaches.

The first group of approaches is the largest one. Decomposition divides the search space into a set of lower-dimensional subspaces by grouping decision variables (or subcomponents), which are expected to be easier to explore independently by an optimization algorithm. For aggregating the whole candidate-solution from subcomponents, the cooperative co-evolution framework is used. Therefore, a decomposition-based approach involves three general components, namely, a decomposition algorithm, a subcomponent optimizer, and a cooperative co-evolution technique. The number of subcomponents and appropriate decomposition depend on the properties of the objective function and are unknown beforehand. Thus, decomposition mechanisms are also a part of the search approach and must be adaptive. Despite the fact that we optimize lower-dimensional subproblems, each decomposition can generate a complex landscape, and the subcomponent optimizer should be also adaptive for demonstrating the high performance for any decomposition. The standard co-evolution framework is also a subject for modification. Nevertheless, the decomposition-based approaches demonstrate high performance for a wide range of LSGO problems.

Learning-based techniques are aimed to identify the interaction of variables and to group them into separable subcomponents. The approaches usually perform well with fully separable and partially non-separable problems only. At the same time, some recent algorithms can also efficiently deal with overlapping components, but still demonstrate poor performance with fully non-separable problems (for example, CC-RDG3, who is the 2019 LSGO competition winner [12]).

Hybrid memetic approaches usually demonstrate high performance for all types of LSGO problems (for example, SHADE-ILS, the 2018 LSGO competition winner [13]). It is worth noting the Multiple Offspring Sampling (MOS) algorithm [14], which was the LSGO competition winner for 5 years (2013–2018). MOS proposes a high-level relay hybrid approach for adaptive switching between global and local search algorithms (one of the modifications uses switching only between multiple local search algorithms).

We will briefly review some state-of-the-art and competition-winner LSGO algorithms for analyzing the general approaches implemented in the algorithms.

### 2.1. Approaches without Problem Decomposition

Dynamic Multi-Swarm Particle Swarm Optimizer (DMS-PSO) [15] is one of the early approaches investigated using the CEC LSGO 2008 benchmark. DMS-PSO uses a multi-population scheme and combines PSO with a modified neighborhood topology [16] and the BFGS Quasi-Newton method for local search. Canonical Differential Evolutionary Particle Swarm Optimization (C-DEEPSO) [17] is based on a combination of DE and PSO algorithms. Variable Mesh Optimization Differential Evolution (VMODE) [18] uses the standard DE as the core optimizer and the population distributed in nodes of a mesh. The mesh nodes can be redistributed for maintaining diversity and for guiding the optimizer to the best-found solutions.

Multi-trajectory Search (MTS) [19] uses a combination of coordinate-wise random searches titled MTS-LS1, MTS-LS2, and MTS-LS3. On each iteration, coordinates are ranked based on the objective improvements, the next step starts with the coordinate that has provided the highest increment of the objective function. Despite the simple idea, MTS demonstrates high performance with LSGO problems and is used as the main local search algorithm in many hybrid memetic approaches.

Iterative Hybridization of Differential Evolution with Local Search (IHDELS) [20] is one of the first competition-winner memetic evolutionary algorithms (the 2nd place in the 2015 IEEE CEC LSGO competition). IHDELS uses self-adaptive DE (SaDE) [21] and two local search algorithms: L-BFGSB [22] and MTS-LS1 [19].

Multiple Offspring Sampling (MOS) in the original paper used a combination of Restart Covariance Matrix Adaptation Evolution Strategy With Increasing Population Size (IPOPCMA-ES) [23] with a restart and variable population size and the standard DE [24]. The 2013 version of MOS [25] uses a hybridization of 3 algorithms: MTS LS1, Solis and Wets, and GA.

Success-History Based Parameter Adaptation for Differential Evolution with Iterative Local Search (SHADE-ILS) [13] is the winner of the 2018 competition. SHADE-ILS combines SHADE [26] for global search, MTS LS1 and L-BFGS-B for local search, and restart strategies.

Hybrid of Minimum Population Search and Covariance Matrix Adaptation Evolution Strategy (MPS-CMA-ES) [27,28] has taken second place in the 2019 competition.

The most recent algorithm selection wizard, titled as automated black-box optimization (ABBO), can select one or several optimization algorithms from a very large number of base algorithms based on some input information about the considered optimization problem [29]. ABBO uses three types of selection techniques: passive algorithm selection, active algorithm selection, and chaining (several algorithms run in turn). ABBO outperforms many state-of-the-art algorithms on LSGO benchmarks.

### 2.2. Decomposition-Based Approaches with Static Grouping

A Cooperative Co-evolutionary approach for Genetic Algorithm (CCGA-1 and CCGA-2) [30] is the first attempt for improving the standard EA (namely, a binary GA) using the coordinate-wise decomposition. In [31,32], CC was implemented for improving evolution programming (Fast Evolutionary Programming with Cooperative Co-evolution, FEPCC). In Cooperative Co-evolutionary Differential Evolution (CCDE) [33], the approach was modified for using subcomponents with many variables. There were proposed two modifications: CCDE-H with 2 subcomponents and CCDE-O with the number of subcomponents equal to the number of variables.

Some more complicated decompositions have been proposed for PSO algorithms. Cooperative Approach to Particle Swarm Optimization (CPSO) [34] performed grouping into k subcomponents (CPSO-Sk) or combined CPSO-Sk with the standard PSO (CPSO-Hk). A similar idea was used in Cooperative Bacterial Foraging Optimization (CBFO) [35] and Cooperative Artificial Bee Colony (CABC) [36]. Both approaches had two modifications: CBFO-S, CBFO-H, CABC-H, and CABC-S.

## 2.3. Decomposition-Based Approaches with Random Grouping

In random grouping approaches, subcomponents vary during the search process. One of the first approaches DECC-G [37] has proposed a combination of random grouping and Self-adaptive Differential Evolution with Neighborhood Search (SaNSDE) [38] that demonstrates high performance for LSGO benchmarks and is still used as a base-line for evaluating and comparing new LSGO approaches.

In Multilevel Cooperative Co-evolution (MLCC) [39], SaNSDE is combined with modified random grouping, which uses a distribution of probabilities for choosing subcomponents from a decomposition pool based on the success of the previous choices. In DECC-ML, a modification of MLCC with a better optimizer for the more frequent random grouping was proposed [40]. Cooperatively Co-evolving Particle Swarms algorithms (CCPSO and CCPSO2) [41,42] use random grouping with PSO. CCPSO2 applies a random search for dynamic regrouping variables. In Cooperative Co-evolution Orthogonal Artificial Bee Colony (CCOABC) [43], random grouping is combined with the ABC algorithm.

Memetic Framework for Solving Large-scale Optimization Problems (MLSHADE-SPA) [44] is a multi-algorithms approach, which iteratively applies Success History-based Differential Evolution with Linear Population Size Reduction (L-SHADE) [45], two self-adaptive DE algorithms, and a modified version of MTS. All algorithms are applied for the whole optimization problem and for subcomponents. MLSHADE-SPA has taken second place in the 2018 IEEE CEC LSGO competition.

## 2.4. Learning-Based Grouping Approaches

The idea behind algorithms of this type is to identify the interaction of decision variables and group them into the same subcomponent. For some separable test problems, the algorithms can identify true subcomponents.

Correlation-based Adaptive Variable Partitioning (CCEA-AVP) [46] evaluates the correlation matrix for the best solutions (in the original algorithm, half of the population is used). Variables for which values of the correlation coefficient are greater than a threshold are placed in one group. In [47], CCEA-AVP uses NSGA-2 as the core optimizer. Contribution Based Cooperative Co-evolution (CBCC) applies SaNSDE with Delta Grouping and Ideal Grouping algorithms [48]. Each subcomponent is optimized using the number of function evaluations based on the improvement of the objective function obtained by this component. The delta grouping approach is also applied in Cooperative Co-evolution with Delta Grouping (DECC-DML) [49]. In Cooperative Co-evolution with Variable Interaction Learning (CCVIL) [50], groups are formed iteratively starting with one-dimensional subcomponents, which are combined if the interaction between them is detected. CCVIL uses JADE [51] for optimizing subcomponents. Dependency Identification with Memetic Algorithm (DIMA) [52] applies the local search algorithm proposed in [53] for detecting the interaction of variables.

Differential grouping is based on the mathematical definition of a partially additively separable function that is used for the identification of the interaction of variables. Cooperative Co-Evolution with Differential Grouping (DECC-DG) [54], Extended Differential Grouping (DECC-XDG) [55], and modified DECC-DG2 [56] use the SaNSDE algorithm for evolving subcomponents. A competitive divide and-conquer algorithm (CC-GDG-CMAES) [57] combines differential grouping with the CMA-ES optimizer. In Differential Grouping with Spectral Clustering (DGSC) [58], SaNSDE is applied for subcomponents discovered using clustering of the identified interactions in variables.

Some original approaches are proposed in Scaling Up Covariance Matrix Adaptation Evolution Strategy (CC-CMA-ES) [59], Cooperative co-evolution with Sensitivity Analysis-based Budget Assignment Strategy (SACC) [60], Bi-space Interactive Cooperative Co-evolutionary Algorithm (BICCA) [61], and Cooperative Co-evolution with Soft Grouping (SGCC) [62]. All approaches, except CC-CMA-ES, use self-adaptive DE algorithms.

A recursive decomposition method (RDG) [63] proposes a new approach for better differential grouping. CC-RDG3 [12] combines CMA-ES with RDG for the efficient identi-

fication of overlapping subcomponents. Authors have shown that CC-RDG3 can greatly improve LSGO algorithms. CC-RDG3 has taken first place in the 2019 IEEE CEC LSGO competition and it is still the leading LSGO approach. An Incremental Recursive Ranking Grouping (IRRG) is one of the recent approaches that uses monotonicity checking for more accurate identification of variable linkages [64]. IRRG requires more fitness function evaluations than RDG3, but never reports false linkages.

### 2.5. LSGO State-of-the-Art Algorithms

We have summarized all approaches mentioned above in Table 1 to highlight their main features, such as the type of decomposition, and the global and local search algorithms used. As we can see from the proposed review, many state-of-the-art LSGO algorithms contain, in different combinations, three main components: problem decomposition with cooperative co-evolution, a global optimizer, and a local search algorithm. The majority of the algorithms apply a self-adaptive DE as the global search technique. In Table 2, all participants of the IEEE CEC LSGO competitions of different years are collected. In the table, one can find out the winners of the competitions and what components (CC, DE, and LS) are implemented in the algorithms (a "plus" sign indicates that the corresponding components are used).

**Table 1.** The summary of the reviewed LSGO approaches.

| Approach | Decomposition Type | Global Search | Local Search |
|---|---|---|---|
| ABBO [29] | No | miniLHSDE | No |
| BICCA [61] | Learning | L-SHADE | No |
| CBCC [48] | Learning | SaNSDE | No |
| CC-CMA-ES [59] | Learning | CMA-ES | No |
| CCDE-H [33] | Static | DE | No |
| CCDE-O [33] | Static | DE | No |
| CCEA-AVP [46] | Learning | NSGA-2 | No |
| CCGA [30] | Static | GA | No |
| CC-GDG-CMAES [57] | Learning | CMA-ES | No |
| CC-RDG3 [12] | Learning | CMA-ES | No |
| CCVIL [50] | Learning | JADE | No |
| C-DEEPSO [17] | Random | EP, PSO, and DE | No |
| DECC-DG [54] | Learning | SaNSDE | No |
| DECC-DG2 [56] | Learning | SaNSDE | No |
| DECC-DML [49] | Learning | SaNSDE | No |
| DECC-G [37] | Random | SaNSDE | No |
| DECC-XDG [55] | Learning | SaNSDE | No |
| DGSC [58] | Learning | SaNSDE | No |
| DIMA [52] | Learning | GA | Self-directed Local Search |
| DMS-PSO [15] | No | PSO | Quasi-Newton method |
| IHDELS [20] | No | SaDE | MTS-LS1, L-BFGS-B |
| IPOPCMA-ES [23] | No | CMA-ES | No |
| IRRG [64] | Learning | CMA-ES | No |
| MLCC [39] | Random | SaNSDE | No |
| MLSHADE-SPA [44] | Random | L-SHADE | MTS-LS1 |
| MOS [14] | No | No | Solis and Wets, MTS-LS1 |
| MOS 2013 [25] | No | GA | Solis Wets, MTS-LS1 |
| MPS-CMA-ES [27] | No | CMA-ES | No |
| MTS [19] | No | No | MTS-LS1, MTS-LS2, MTS-LS3 |
| SACC [60] | Learning | SaNSDE | No |
| SGCC [62] | Learning | SaNSDE | No |
| SHADE-ILS [13] | No | SHADE | MTS-LS1, L-BFGS-B |
| VMODE [18] | Static | DE | No |

**Table 2.** LSGO state-of-the-art algorithms.

| Algorithm | Year | Winner | 2nd | 3rd | CC | DE | LS |
|---|---|---|---|---|---|---|---|
| CC-CMA-ES | 2015 | | | 2015 | + | − | − |
| DEEPSO | 2015 | | | | - | + | − |
| IHDELS | 2015 | | 2015 | | − | + | + |
| SACC | 2015 | | | | + | + | + |
| VMODE | 2015 | | | | − | + | − |
| BICCA | 2018 | | | | + | − | − |
| MPS | 2019 | | 2019 | | − | − | − |
| SGCC | 2019 | | | 2019 | + | + | − |
| DECC-G | 2015, 2018 | | | | + | + | − |
| MOS | 2015, 2018 | 2015 | | 2018 | − | + | + |
| MLSHADE-SPA | 2018 | | 2018 | | − | + | + |
| SHADE-ILS | 2018 | 2018 | | | − | + | + |
| CC-RDG3 | 2019 | 2019 | | | + | − | − |
| DGSC | 2019 | | | | + | + | − |

As we can see from Table 2, 10 of 14 participants use DE as a core global optimizer, 7 algorithms apply problem decomposition with CC, and 5 algorithms are memetic. All winners use DE and all winners, except CC-RDG3, use local search. The current leader, CC-RDG3, applies problem decomposition with CC. From a historical perspective, we can notice that leading approaches improve global and local algorithms and develop more advanced frameworks for problem decomposition and adaptive control of the interaction of global and local search. This fact motivates us to design new approaches that combine all 3 components.

## 3. The Proposed Approach

As we can see from the review in the previous section, the majority of state-of-the-art approaches use CC. At the same time, many CC approaches apply an additional learning stage before the main subcomponents' optimization stage. The learning stage is used for identifying interconnected and independent variables. The identification of non-separable groups of variables usually takes a sufficiently large number of function evaluations (FEVs), which could be utilized for the main optimization process. However, the finding of all non-separable groups does not guarantee the high efficiency of solving the obtained optimization sub-problems.

The proposed approach uses an adaptive change of the number of subcomponents, which leads to a dynamic redistribution of the use of computational resources. The set of values of the number of subcomponents is predefined and it is a parameter of the algorithm, which can be set based on the limitations of FEVs. As it was shown in [65,66], it is better to use different decompositions for different stages of the optimization process while exploring different regions of the search space instead of using the only decomposition even it is correct. We have discovered that, in general, an optimizer better operates small subcomponents at the early stages and the whole solution vector at the final stage, when a basin of a global optimum is discovered. Additionally, the way of adaptive changing the number of subcomponents can vary for different types of LSGO problems.

We will use the following algorithm for adaptive change of the number of subcomponents. We will run many optimizers, which use decompositions with subcomponents of different sizes. Each algorithm uses its number of FEVs based on its success in the previous generations. Thus, we dynamically redistribute resources in favor of a more preferable decomposition variant.

A set of $M$ values of the number of subcomponents is defined as $\{CC_1, CC_2, \ldots, CC_M\}$, where all elements should be different, i.e., $CC_1 \neq CC_2 \neq \ldots \neq CC_M$. At the initialization stage, for each of M algorithms, we assign equal resources defined as the number of generations ($G_i, i = 1, \ldots, M$). After all resources are exhausted by

algorithms, we start a new cycle by redistributing the resources. In each cycle, algorithms are applied consequently in random order.

At the end of the run of each algorithm, we evaluate the improvement rate (2):

$$improvment\_rate_i = \frac{(best\_found_{before} - best\_found_{after})}{best\_found_{after}}, \tag{2}$$

here $best\_found_{before}$ is the best-found solution before the run, and $best\_found_{after}$ is the best-found solution after the run, and $i = 1, \ldots, M$.

After each cycle, all algorithms are ranked by their improvement rates. The best algorithm increases its resource by $G_{win}$ generations, which is a sum of $G_{lose}$ generations subtracted from resources of all the rest algorithms. For all algorithms, we define $G_{min}$ for preventing the situation when the current-winner algorithm takes all resources and eliminates all other participants.

In the proposed approach, we will run the MTS-LS1 algorithm after CC-based SHADE. Usually, MTS-LS1 can find a new best-found solution that becomes far from other individuals in the population. In this case, SHADE cannot improve the best-found solution for a long time but improves the average fitness in the population. Criterion (2) becomes insensitive to differences in the behavior of algorithms if the improving rate is calculated using the best-found solution. To overcome this difficulty, we will calculate the improving rate using the median fitness before and after an algorithm run using (3) instead of (2).

$$improving\_rate_i = \frac{medianFitness_{before} - medianFitness_{after}}{medianFitness_{after}}, \tag{3}$$

here *medianFitness* is the median fitness of individuals in the population, and $i = 1, \ldots, M$.

We will use the following approach for changing the number of generations assigned for each algorithm in the cooperation (4)–(8):

$$IR = \left\{ i : improvment\_rate_i = \max_{j=1,\ldots,M} \{improvment\_rate_j\} \right\} \tag{4}$$

$$NI = |IR| \tag{5}$$

$$pool = \sum_{j=1}^{M} \begin{cases} G_{lose}, & \text{if } (G_i - G_{lose} \geq G_{min}) \wedge i \notin IR \\ 0, & \text{otherwise} \end{cases} \tag{6}$$

$$G_{win} = \left\lfloor \frac{pool}{NI} \right\rfloor \tag{7}$$

$$G_i = \begin{cases} G_i + G_{win}, & \text{if } i \in IR \\ G_i - G_{lose}, & \text{if } (G_i - G_{lose} \geq G_{min}) \wedge i \notin IR \\ G_{min}, & \text{if } (G_i - G_{lose} < G_{min}) \wedge i \notin IR \end{cases} \tag{8}$$

here *IR* is a set of indexes of algorithms with the best improving rate, *NI* is the number of algorithms with the best improving rate, *pool* is a pool of resources for redistribution, and $i = 1, \ldots, M$.

We will use SHADE as a core optimizer for subcomponents in CC-based algorithms. In our review, we have shown that almost all competition winners and state-of-the-art algorithms use one of the modifications of DE. The main benefit of the SHADE algorithm in solving "black-box" optimization problems is that it has only two control parameters, which can be automatically tuned during the optimization process [67].

The main parameters of DE (scale factor *F* and crossover rate *Cr*) in SHADE are self-configuring. SHADE uses a historical memory, which contains *H* pairs of the parameters from the previous generations. A mutant vector is created using a random pair of the

parameters from the historical memory. When applying SHADE with CC, we will use specific parameters *Cr* and *F* for each subcomponent.

SHADE, like many other DE algorithms, uses an external archive for saving some promising solutions from the previous generations. SHADE records the parameter values and the corresponding function increments when a better solution is found. After each generation, SHADE calculates new values of the control parameters using the weighted Lehmer mean [67]. New calculated values of *Cr* and *F* are placed in the historical memory.

SHADE uses the *current-to-pbest/1* mutation scheme. The archived solutions can be chosen and reused at the mutation stage for maintaining the population diversity.

Our experimental results have shown that the use of independent populations and archives for each of the algorithms does not increase the overall performance of the proposed approach. In this work, all algorithms in the cooperation use the same population and archive.

One of the important control parameters of EA-based algorithms is the population size. A large population is more preferable at the exploration stage, when the algorithm converges, it loses the population diversity and the population size can be increased. If the variance of coordinates is high (individuals are well distributed in the search space), we reduce the population size and drop out randomly chosen solutions except the best one. We will use an adaptive size of the population based on the analysis of the diversity. The following diversity measure (9) is used [68]:

$$DI = \sqrt{\frac{1}{NP} \sum_{i=1}^{NP} \sum_{j=1}^{n} \left(x_{ij} - \overline{x}_j\right)^2}, k = 1, \ldots, M, \tag{9}$$

here *NP* is the population size, *n* is the dimensionality of the objective function, $\overline{x}_j$ is the average value of the *j*-th variable of all individuals in the population.

After each cycle, we define a new population size for each algorithm using (10)–(13).

$$RD = \frac{DI}{DI_{init}} \tag{10}$$

$$RFES = \frac{FEVs}{maxFEVs} \tag{11}$$

$$rRD = 1 - \frac{RFES}{0.9} \tag{12}$$

$$NP = \begin{cases} NP + 1, & \text{if } (NP + 1 \leq maxNP) \wedge (RD < 0.9 \cdot rRD) \\ NP - 1, & \text{if } (NP - 1 \geq minNP) \wedge (RD > 1.1 \cdot rRD) \\ NP, & \text{otherwise} \end{cases} \tag{13}$$

here *RD* is relative diversity, *RFES* is a relative spend of the FEV budget (*maxFEVs*), *rRD* is the required value of *RD*, *minNP*, and *maxNP* are low and upper bounds for the population size.

The relationship between *RD* and *RFES* is presented in Figure 1.

If the variance of coordinates is high (individuals are well distributed in the search space), we reduce the population size and drop out randomly chosen solutions except the best one. If the variance is low (individuals are concentrated in some region of the search space), we increase the population size by adding new random solutions. The approach tries to keep relative diversity close to *rRD*, which linearly decreases with spending FEVs.

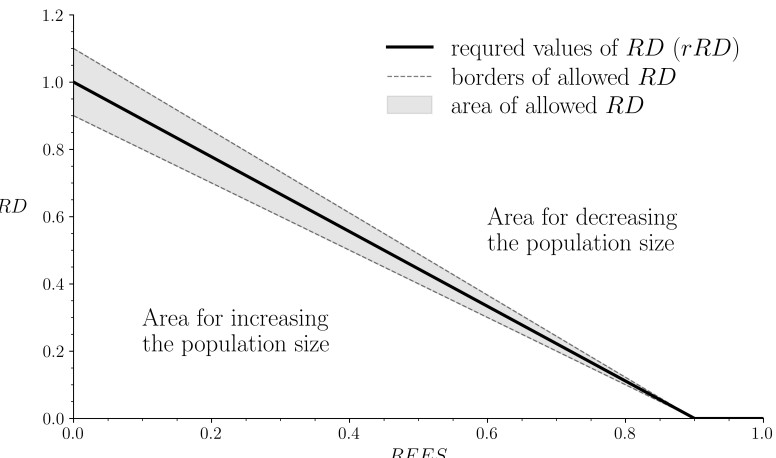

**Figure 1.** Diversity-based mechanism of population size adaptation.

The proposed above ideas are implemented in our new algorithm titled COSACC-LS1. One of the hyper-parameters of COSACC-LS1 is the number of algorithms with different subcomponents ($M$). Because of the high computational cost of LSGO experiments, in this research we have tried $M = 3$ and the following combinations of the number of subcomponents: $\{1,2,4\}$, $\{1,2,8\}$, $\{1,2,10\}$, $\{1,4,8\}$, $\{1,4,10\}$, $\{1,8,10\}$, $\{2,4,8\}$, and $\{2,4,10\}$. Thereafter, we will use the notation "COSACC-LS1 $\{x,y,z\}$", where $x$, $y$, and $z$ stands for the number of subcomponents, which are used in three DE algorithms: CC-SHADE(x), CC-SHADE(y), and CC-SHADE(z).

We have tried different mutation schemes and have obtained that the best performance of COSACC-LS1 is reached using the following scheme (14):

$$u_i = x_i + F_i \cdot \left( x_{pbest} - x_i \right) + F_i \cdot (x_t - x_r), i = 1, \ldots, NP, \qquad (14)$$

here, $u_i$ is a mutant vector, $F_i$ is the scale factor, $x_{pbest}$ is a random solution chosen from the $p$ best solutions, $x_t$ is an individual chosen using the tournament selection from the population (the tournament size is 2), $x_r$ is a random solution chosen from the union of the current population and the archive, and all solutions chosen for performing mutation must be different, i.e., $i \neq pbest \neq t \neq r$.

The size of the archive is set two times larger than the initial population size. The size of historical memory in SHADE is set to 6 (the value is defined using grid search).

We have chosen MTS-LS1 for implementing local search in COSACC, because it demonstrates high performance in solving LSGO problems both alone and when applied with a global search algorithm [19]. We use the following settings for MTS-LS1. The maximum number of FEVs is 25000 (the value is defined by numerical experiments). MTS-LS1 searches along each $i$-th coordinate using the search range $SR[i]$. The initialization of $SR[i]$ is the same as in the original MTS: $(SR[i] = (b - a) \cdot 0.4)$, where $[a, b]$ is low and high bounds for the i-th variable. If a better solution is not found using the current value of $SR[i]$, it is reduced $(SR[i] = SR[i]/2)$. If $SR[i]$ becomes less than 1E-18 (the original threshold was 1E-15), the value is reinitialized.

MTS-LS1 is applied after each main cycle starting with the current best-found solution until maximum FEVs are reached.

The initial number of generations for all algorithms is 15, the minimum value is 5, respectively. After a cycle, we will add $(M - 1)$ generation to $G$ for the algorithm with the highest improving rate. All other algorithms will reduce the number of generations by one. The initial population size is 100, minimum and maximum values are 25 and 200, respectively. After the algorithm spends 90% of its computational resource, the population size is set to its minimum value as proposed in [68] (in this work the value is 25).

The whole implementation scheme for the proposed approach is presented using pseudocode in Algorithm 1.

---

**Algorithm 1** The general scheme of COSACC-LS1

---

**Require:** The number of algorithms $M$ in CC , the number of subcomponents for each algorithm, $n$, $NP$, $minNP$, $maxNP$, $G_{init}$, $G_{lose}$, $G_{min}$, $maxFEVs$.
**Ensure:**
  $population \leftarrow RandomPopulation(n, NP)$
  $DI_{init} \leftarrow CalculateDiversity(population)$            ▷ Using Equation (9)
  **for all** $i = 1, \ldots, M$ **do**
    $G_i \leftarrow G_{init}$
  **end for**
  **while** $FEVs < maxFEVs$ **do**
    **for all** $i \in RandomPermutation(1, \ldots, M)$ **do**
    $medianFitness\_before \leftarrow GetMedianFitness(population)$
    **for** $g \leftarrow 1, G_i$ **do**
      $best\_found \leftarrow CC\text{-}SHADE(population, NP, i)$
      $RD \leftarrow EvalRD(DI_{init}, population, NP)$         ▷ Equation (10)
      $NP \leftarrow EvalPopsize(RD, maxFEVs, NP, maxNP)$    ▷ Equation (13)
    **end for**
    $medianFitness\_after \leftarrow GetMedianFitness(population)$
    $improving\_rate_i \leftarrow \frac{medianFitness\_before - medianFitness\_after}{medianFitness\_after}$
    **end for**
    **for all** $i = 1, \ldots, M$ **do**
      $G_i \leftarrow EvalNumGenerations(improving\_rate_i)$       ▷ Equation (8)
    **end for**
    $best\_found \leftarrow GetBestFound(population)$
    $best\_found \leftarrow MTS\text{-}LS1(best\_found)$
  **end while**

---

## 4. Experimental Setups and Implementation

We have investigated the performance of COSACC-LS1 and have compared the results with other state-of-the-art approaches using the actual LSGO benchmark, proposed at the special session of IEEE Congress on Evolutionary Computation in 2013 [5]. The benchmark proposes 15 "black-box" real-valued LSGO problems. There are 4 types of problems, namely fully-separable functions (F1–F3), partially separable functions (F4–F11), functions with overlapping subcomponents (F12–F14), and fully-nonseparable functions (F15). The functions have many features, which complicate solving the problems using standard EAs and other metaheuristics. Some of the features are non-uniform subcomponent sizes, imbalance in the contribution of subcomponents, overlapping subcomponents, transformations to the base functions, ill-conditioning, symmetry breaking, and irregularities [48,69].

The performance measure for LSGO algorithms is the error of the best-found solution averaged over 25 independent runs. The error is an absolute difference between the best-found solution and the true value of a global optimum. The maximum FEVs in a run is 3.0E+06. Based on the benchmark rules, the following additional data is collected: for each problem, the best-found fitness values averaged over 25 runs are saved after 1.2E+05, 6.0E+05, and 3.0E+06 FEVs. We also will estimate the variance of the results using the best, median, worst, mean, and standard deviation of the results.

Authors of the LSGO CEC 2013 benchmark propose software implementation using C++, Java, and Python programming languages. For a fair comparison of the results with other state-of-the-art algorithms, the Toolkit for Automatic Comparison of Optimizers (TACO) [6,70] is used. TACO is an online database, which proposes the automatic comparison of the results uploaded by users with the results of selected LSGO algorithms stored in the database. TACO presents reports of the results of ranking the selected algorithms

based on the Formula 1 ranking system. The ranking is presented for the whole benchmark and each of the 4 types of problems.

Experimental analysis of new LSGO approaches is very expensive in terms of computational time. For all computational experiments, the proposed approach has been implemented using C++. The C++ language usually demonstrates higher computing speed and has wide possibilities for parallelization using many computers with many CPU cores. We have designed and assembled our computational cluster based on 8 AMD Ryzen Pro CPUs, which, in total, supply 128 threads for parallel computing. The MPICH2 (Message Passing Interface Chameleon) framework for connecting all PCs in the cluster is used. The Master–Slave communication scheme with the queue is applied. The operating system is Ubuntu LTS 20.04. One series of experiments using the LSGO benchmark using the cluster takes about 2 h compared to 265 h when using a single computer with regular sequential computing. The source codes and additional information on our cluster are available on https://github.com/VakhninAleksei/COSACC-LS1 (accessed on 01 September 2022).

## 5. The Experimental Results

The results of evaluating COSACC-LS1 with the best configuration {1, 2, 4} on the IEEE CEC LSGO benchmark are presented in Table 3. The results contain the best, median, worst, mean, and standard deviation values of the best-found solutions from 25 independent runs after 1.2E+05, 6.0E+05, and 3.0E+06 FEVs (following the benchmark rules).

**Table 3.** The experimental results on the IEEE CEC 2013 LSGO benchmark.

| Problems: | | F1 | F2 | F3 | F4 | F5 | F6 | F7 | F8 |
|---|---|---|---|---|---|---|---|---|---|
| | Best | 2.82E-06 | 1.02E+03 | 2.00E+01 | 1.28E+10 | 1.55E+06 | 1.04E+06 | 1.14E+09 | 6.24E+14 |
| | Median | 5.13E-06 | 1.14E+03 | 2.00E+01 | 1.23E+11 | 3.06E+06 | 1.05E+06 | 2.33E+09 | 2.05E+15 |
| 1.20E+05 | Worst | 1.05E-05 | 1.29E+03 | 2.00E+01 | 2.52E+11 | 4.88E+06 | 1.06E+06 | 4.86E+09 | 1.24E+16 |
| | Mean | 5.68E-06 | 1.14E+03 | 2.00E+01 | 1.32E+11 | 3.34E+06 | 1.05E+06 | 2.51E+09 | 2.92E+15 |
| | StDev | 2.10E-06 | 7.60E+01 | 1.91E-04 | 5.99E+10 | 9.76E+05 | 4.11E+03 | 9.57E+08 | 2.48E+15 |
| | Best | 0.00E+00 | 1.00E+03 | 2.00E+01 | 2.33E+09 | 9.08E+05 | 1.04E+06 | 2.62E+07 | 5.97E+13 |
| | Median | 0.00E+00 | 1.12E+03 | 2.00E+01 | 8.63E+09 | 1.07E+06 | 1.04E+06 | 6.26E+07 | 2.94E+14 |
| 6.00E+05 | Worst | 6.53E-24 | 1.24E+03 | 2.00E+01 | 3.95E+10 | 1.76E+06 | 1.05E+06 | 1.50E+08 | 7.11E+14 |
| | Mean | 2.61E-25 | 1.12E+03 | 2.00E+01 | 1.34E+10 | 1.13E+06 | 1.05E+06 | 6.74E+07 | 3.04E+14 |
| | StDev | 1.31E-24 | 7.42E+01 | 1.91E-04 | 1.09E+10 | 2.15E+05 | 3.34E+03 | 2.94E+07 | 1.69E+14 |
| | Best | 0.00E+00 | 1.00E+03 | 2.00E+01 | 1.21E+08 | 9.08E+05 | 1.04E+06 | 2.00E+02 | 1.36E+13 |
| | Median | 0.00E+00 | 1.11E+03 | 2.00E+01 | 1.27E+09 | 1.07E+06 | 1.04E+06 | 1.00E+04 | 6.58E+13 |
| 3.00E+06 | Worst | 0.00E+00 | 1.22E+03 | 2.00E+01 | 7.76E+09 | 1.76E+06 | 1.05E+06 | 1.80E+05 | 2.99E+14 |
| | Mean | 0.00E+00 | 1.11E+03 | 2.00E+01 | 2.17E+09 | 1.13E+06 | 1.04E+06 | 3.16E+04 | 8.02E+13 |
| | StDev | 0.00E+00 | 7.29E+01 | 1.57E-04 | 2.07E+09 | 2.15E+05 | 1.80E+03 | 5.38E+04 | 6.86E+13 |
| | | F9 | F10 | F11 | F12 | F13 | F14 | F15 | |
| | Best | 1.78E+08 | 9.26E+07 | 2.35E+10 | 9.62E+02 | 3.22E+09 | 1.62E+11 | 6.37E+07 | |
| | Median | 3.36E+08 | 9.38E+07 | 1.15E+11 | 1.95E+03 | 2.66E+10 | 3.55E+11 | 1.10E+08 | |
| 1.20E+05 | Worst | 4.30E+08 | 9.47E+07 | 3.11E+11 | 8.24E+03 | 4.65E+10 | 6.76E+11 | 2.21E+08 | |
| | Mean | 3.22E+08 | 9.38E+07 | 1.21E+11 | 2.48E+03 | 2.84E+10 | 3.87E+11 | 1.16E+08 | |
| | StDev | 6.69E+07 | 5.68E+05 | 8.24E+10 | 1.92E+03 | 8.76E+09 | 1.49E+11 | 3.84E+07 | |
| | Best | 8.42E+07 | 9.23E+07 | 7.22E+08 | 1.43E+02 | 1.09E+08 | 7.61E+08 | 1.11E+07 | |
| | Median | 1.23E+08 | 9.32E+07 | 1.34E+09 | 7.37E+02 | 1.90E+09 | 6.60E+09 | 1.54E+07 | |
| 6.00E+05 | Worst | 1.62E+08 | 9.38E+07 | 1.18E+10 | 1.62E+03 | 3.09E+09 | 2.68E+10 | 2.86E+07 | |
| | Mean | 1.25E+08 | 9.31E+07 | 1.71E+09 | 7.13E+02 | 1.97E+09 | 8.37E+09 | 1.67E+07 | |
| | StDev | 2.09E+07 | 4.27E+05 | 2.15E+09 | 3.23E+02 | 6.42E+08 | 6.58E+09 | 4.69E+06 | |
| | Best | 8.42E+07 | 9.23E+07 | 1.34E+06 | 7.72E-09 | 5.52E+04 | 6.53E+06 | 1.10E+06 | |
| | Median | 1.23E+08 | 9.27E+07 | 2.69E+06 | 1.20E+01 | 1.42E+06 | 9.11E+06 | 1.48E+06 | |
| 3.00E+06 | Worst | 1.62E+08 | 9.32E+07 | 3.70E+07 | 2.58E+02 | 2.60E+06 | 1.61E+07 | 2.26E+06 | |
| | Mean | 1.25E+08 | 9.27E+07 | 6.74E+06 | 5.02E+01 | 1.42E+06 | 9.25E+06 | 1.52E+06 | |
| | StDev | 2.10E+07 | 2.17E+05 | 1.06E+07 | 7.25E+01 | 6.37E+05 | 2.07E+06 | 2.94E+05 | |

At first, the results of COSACC-LS1 have been compered with its component algorithms, COSACC and LS1, to prove the benefits of their cooperation. Both component algorithms have been evaluated using their best settings obtained with the grid search. All

comparisons have been performed using the median of the best-found solutions in the runs after spending the full FEVs budget. Table 4 contains the medians and the results of the Mann–Whitney–Wilcoxon (MWW) tests and ranking. High values of ranks are better. When the difference in the results is not statistically significant, algorithms share ranks. The average ranks are presented in Figure 2.

**Table 4.** The comparison of algorithms.

| Problems: | F1 | F2 | F3 | F4 | F5 | F6 | F7 | F8 |
|---|---|---|---|---|---|---|---|---|
| | | | | The median of the best-found solution | | | | |
| COSACC (A1) | 2.22E-14 | 6.70E+03 | 2.02E+01 | 3.02E+09 | 1.31E+06 | 1.06E+06 | 7.46E+05 | 3.04E+13 |
| LS1 (A2) | 0.00E+00 | 1.10E+03 | 2.00E+01 | 1.21E+11 | 2.05E+07 | 1.05E+06 | 1.44E+09 | 1.07E+16 |
| COSACC-LS1 (A3) | 0.00E+00 | 1.11E+03 | 2.00E+01 | 2.17E+09 | 1.13E+06 | 1.04E+06 | 3.16E+04 | 8.02E+13 |
| | | | | The MWW test | | | | |
| A1 vs. A3, *p*-value | 5.96E-08 | 5.96E-08 | 5.96E-08 | 1.34E-01 | 2.36E-02 | 5.96E-08 | 5.96E-08 | 1.40E-04 |
| A1 vs. A2, *p*-value | 5.96E-08 | 5.96E-08 | 5.96E-08 | 5.96E-08 | 5.96E-08 | 5.96E-08 | 5.96E-08 | 5.96E-08 |
| A2 vs. A3, *p*-value | 0.00E+00 | 3.25E-01 | 5.49E-02 | 5.96E-08 | 5.96E-08 | 7.50E-05 | 5.96E-08 | 5.96E-08 |
| | | | | The ranking of algorithms | | | | |
| COSACC (A1) | 1 | 1 | 1 | 2.5 | 2 | 1 | 2 | 3 |
| LS1 (A2) | 2.5 | 2.5 | 2.5 | 1 | 1 | 2 | 1 | 1 |
| COSACC-LS1 (A3) | 2.5 | 2.5 | 2.5 | 2.5 | 3 | 3 | 3 | 2 |
| | **F9** | **F10** | **F11** | **F12** | **F13** | **F14** | **F15** | |
| | | | | The median of the best-found solution | | | | |
| COSACC (A1) | 1.38E+08 | 9.39E+07 | 1.18E+07 | 1.85E+03 | 8.23E+06 | 2.07E+07 | 4.51E+05 | |
| LS1 (A2) | 1.76E+09 | 9.42E+07 | 2.21E+11 | 1.23E+03 | 1.35E+10 | 3.20E+11 | 7.59E+07 | |
| COSACC-LS1 (A3) | 1.25E+08 | 9.27E+07 | 6.74E+06 | 5.02E+01 | 1.42E+06 | 9.25E+06 | 1.52E+06 | |
| | | | | The MWW test | | | | |
| A1 vs. A3, *p*-value | 9.03E-02 | 5.96E-08 | 1.05E-02 | 5.96E-08 | 5.96E-08 | 2.98E-07 | 5.96E-08 | |
| A1 vs. A2, *p*-value | 5.96E-08 | 7.55E-02 | 5.96E-08 | 1.36E-02 | 5.96E-08 | 5.96E-08 | 5.96E-08 | |
| A2 vs. A3, *p*-value | 5.96E-08 | 5.96E-08 | 5.96E-08 | 5.96E-08 | 5.96E-08 | 5.96E-08 | 5.96E-08 | |
| | | | | The ranking of algorithms | | | | |
| COSACC (A1) | 2.5 | 1.5 | 2 | 1 | 2 | 2 | 3 | |
| LS1 (A2) | 1 | 1.5 | 1 | 2 | 1 | 1 | 1 | |
| COSACC-LS1 (A3) | 2.5 | 3 | 3 | 3 | 3 | 3 | 2 | |

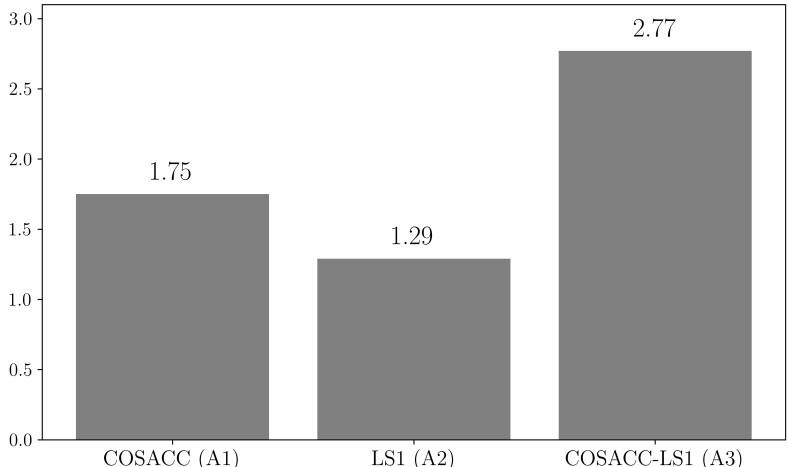

**Figure 2.** Average ranks of COSACC, LS1, and COSACC-LS1 algorithms.

As we can see from the results, COSACC-LS1 has won 8 times, 5 times has shared first place with a component algorithm, and 2 times has taken second place. On easy separable problems (F1–F6), single COSACC yields to both algorithms, because it spends the budget for exploration of the search space while LS1 greedy converges to an optimum. On average,

COSACC-LS1 obtains the best ranks, thus, in the case of black-box LSGO problems, the choice of the hybrid approach is preferable.

The following statistical data for each benchmark problem collected during independent runs of COSACC-LS1 have been visualized: convergence, dynamics of the population size, and redistribution of the computational resources for algorithms with the different number of subcomponents. Each plot presents the mean and standard deviation of 25 runs. The whole set of plots is presented in Appendix A (Figures A1–A15).

## 6. Discussion

In this section, we have analyzed 3 general situations in the algorithm behavior based on plots for F3, F8, and F10 problems.

LSGO problems are hard for many search techniques when they optimize the complete solution vector, and the problem decomposition can ease this issue. In our previous studies, we have discovered that the cooperation of multiple algorithms with a different number of subcomponents usually demonstrates the following usage of decompositions. At the initial generations, the best performance is obtained using many subcomponents of small sizes. Such component-wise search performs the exploration strategy. After that, the approach usually chooses algorithms with a smaller number of subcomponents and at the final generations, it optimizes the complete solution vector. Optimization without decomposition tries to improve the final solution and performs the exploitation strategy [65]. A similar behavior we can see for COSACC-LS1.

Figure A3 (see Appendix A) shows the dynamics of the algorithms on the F3 problem. F3 is a fully separable problem based on the Ackley function. At the same time, the problem is one of the hardest in the benchmark. The basin of global optimum is narrow. F3 has a huge number of local optima with almost the same values, which cover most of the search space.

As we can see in Figure A3a, the algorithm demonstrates fast convergence at the initial generations and after that, there are no significant improvements in the best-found value. The population size at the initial generations is big because the population diversity ($DI$) becomes less than the required relative diversity ($rRD$). This is the result of the fast convergence, and the algorithm tries to increase the population size up to the threshold value (Figure A3b). When the algorithm falls into stagnation, individuals save their positions, and the diversity becomes greater than $rRD$, thus the population size decreases. As we can see from low STD values, the situation is repeated in every run. The resource redistribution plots (Figure A3c) show that at the initial generations the algorithm prefers to use many subcomponents, but when it falls into stagnation, the algorithm takes this as the end of the exploration and gives resources for optimizing the complete solution vector.

Figure A8 shows the dynamics of the algorithms on the F8 problem, which is a combination of 20 non-separable shifted and rotated elliptic functions. The problem is assumed to be a good test function for decomposition-based approaches, but each subcomponent is a hard optimization problem, which is non-separable and has local irregularities.

As we can see in Figure A8a, the proposed approach demonstrates good convergence at the beginning of the optimization process and then stagnates. Figure A8b shows that the fast convergence leads to a loss in diversity ($DI$) and the algorithm increases the population size until 50% of FEVs is reached. In the middle of the budget spend, individuals have almost the same fitness values and do not improve the best-found value (plateau area in Figure A8b). Finally, the diversity ($DI$) becomes less than the required relative diversity ($rRD$) and the population size decreases. In contrast with the results on F3, before the algorithm falls into stagnation, the fast improvements in the objective lead to an increase in the population size for preventing local convergence.

Figure A8c shows that the algorithm distributes computational resources almost in equal portions on average. We can see an example of the true cooperative search when all

component algorithms support each other. The standard deviation of the redistribution is high because the algorithm permanently adapts $G_i$ values in the run.

Figure A10 presents the convergence on the F10 problem. F10 is a combination of 20 non-separable shifted and rotated Ackey's functions. As it was said previously, the Ackley function is one of the hardest in the benchmark and all Ackley-based problems are also very challenging tasks for LSGO approaches.

As we can in Figure A10a, the algorithm improves the fitness value permanently during the run. At the same time, the relative value of the improvements is low, and coordinates of individuals remain almost the same. The $DI$ value becomes less than $rRD$ at the early generations and the algorithm decreases the population size (Figure A10b). As we have mentioned previously, slow convergence and stagnation usually are the result of the end of the exploration stage and the algorithm prefers to optimize the complete solution vector instead of decomposition-based subcomponents. As we can see in Figure A10c, COSACC-LS1 gives all resources to the component algorithm with no decomposition.

Here it should be noted that in all experiments all component algorithms have a minimum guarantee amount of the computational resource. Even when one of the algorithms is leading, this can still be the result of the cooperation of multiple decompositions, and their small contribution essentially increases the performance of the leading algorithm.

As we can see from the presented convergence plots, COSACC-LS1 demonstrates the self-configuration capability. The approach can adaptively select the best decomposition option using redistribution of the computational resource. Different behavior for different LSGO problems ensures that COSACC-LS1 adapts to the topology of the given objective function. Another feature of the proposed approach is the adaptive control of the population size that maintains the population diversity and prevents the premature convergence.

Finally, the results of COSACC-LS1 have been compared with state-of-the-art approaches using the TACO online database. For the comparison, we have selected all algorithms, which were winners and prize-winners of all previous IEEE CEC LSGO competitions: CC-CMA-ES, CC-RDG3, IHDELS, MLSHADE-SPA, MOS, MPS, SGCC, and SHADEILS (see Table 2). Additionally, we have added DECC-G as it is used as a baseline in the majority of studies and experimental comparisons. Table 5 and Figure 3 show the results of the comparison. For all algorithms, we can see the sum of scores obtained on all benchmark problems and the sum of scores for each type of LSGO problems. The following notation for classes of LSGO problems is used: non-separable functions (Class 1), overlapping functions (Class 2), functions with no separable subcomponents (Class 3), functions with a separable subcomponent (Class 4), and fully-separable functions (Class 5).

**Table 5.** Comparison of state-of-the-art algorithms.

|  | Class 1 | Class 2 | Class 3 | Class 4 | Class 5 | Sum | Mean | Std |
|---|---|---|---|---|---|---|---|---|
| SHADEILS | 25 | 61 | 49 | 57 | 33 | 225 | 45 | 15.49 |
| COSACC-LS1 | 15 | 58 | 48 | 45 | 43 | 209 | 41.8 | 16.05 |
| CC-RDG3 | 12 | 35 | 76 | 68 | 14 | 205 | 41 | 29.83 |
| MLSHADE-SPA | 4 | 36 | 56 | 48 | 60 | 204 | 40.8 | 22.52 |
| MOS | 10 | 32 | 37 | 46 | 58 | 183 | 36.6 | 17.85 |
| IHDELS | 8 | 32 | 33 | 28 | 27 | 128 | 25.6 | 10.16 |
| MPS | 6 | 7 | 28 | 46 | 13 | 100 | 20 | 16.99 |
| SGCC | 18 | 20 | 33 | 23 | 4 | 98 | 19.6 | 10.45 |
| CC-CMA-ES | 2 | 14 | 32 | 19 | 28 | 95 | 19 | 11.87 |
| DECC-G | 1 | 8 | 15 | 24 | 30 | 78 | 15.6 | 11.72 |

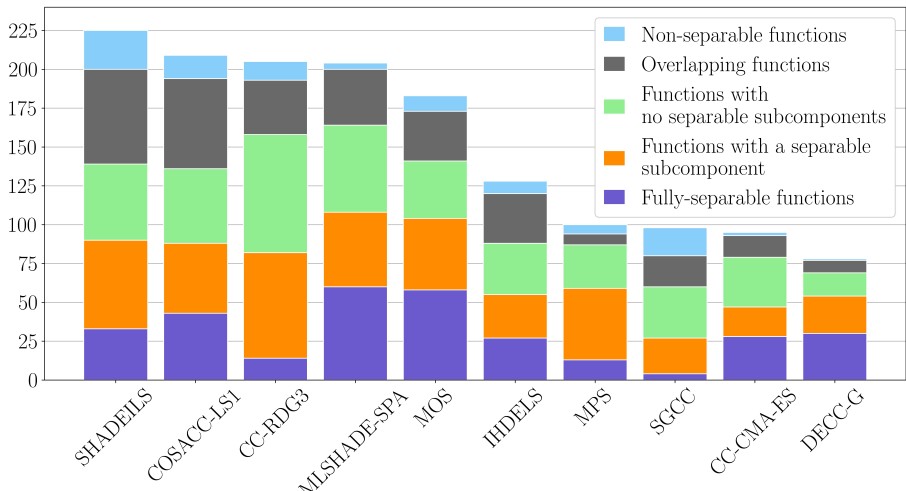

**Figure 3.** Summary scores of state-of-the-art algorithms.

The LSGO benchmark contains only one fully non-separable problem and three problems with overlapping components, which are the hardest problems. At the same time, an algorithm can obtain high summary scores if it has high scores for the type of LSGO problem, which contains many problems.

As we can see from the comparison, CC-RDG3, SHADE-SPA, and MOS have average scores for non-separable and overlapping problems but perform well for other types of LSGO problems. SGCC is the second-best in solving non-separable problems and yields in solving all the rest. SHADEILS is still the competition winner, but it demonstrates low performance when solving fully separable problems because it does not use any decomposition approach that can improve the results for this type.

To better investigate the results for each class of LSGO problems, we have adjusted the given scores by the number of problems of each class. Table 6 shows the results adjusted for the number of problems in each class. Figure 4 demonstrates the variance of scores for the 5 best algorithms. MLSHADE-SPA and MOS obtain high scores for fully-separable functions (outliers in Figure 4), although the results for all other classes have low variance, they are below median values of leading approaches. Median values of SHADEILS and COSACC-LS1 are close, but, as we can see, COSACC-LS1 has less variance thus the results are more stable. We can see that the variance in SHADEILS is towards larger ranks, at the same time this is true only with this benchmark set, because the approach is fine-tuned for the benchmark.

**Table 6.** Comparison of state-of-the-art algorithms.

|  | Class 1 | Class 2 | Class 3 | Class 4 | Class 5 | Sum | Mean | Std |
|---|---|---|---|---|---|---|---|---|
| SHADEILS | 25 | 20.33 | 12.25 | 14.25 | 11 | 82.83 | 16.57 | 5.92 |
| COSACC-LS1 | 15 | 19.33 | 12 | 11.25 | 14.33 | 71.92 | 14.38 | 3.18 |
| CC-RDG3 | 12 | 11.67 | 19 | 17 | 4.67 | 64.33 | 12.87 | 5.57 |
| MLSHADE-SPA | 4 | 12 | 14 | 12 | 20 | 62 | 12.4 | 5.73 |
| MOS | 10 | 10.67 | 9.25 | 11.5 | 19.33 | 60.75 | 12.15 | 4.1 |
| IHDELS | 8 | 10.67 | 8.25 | 7 | 9 | 42.92 | 8.58 | 1.37 |
| SGCC | 18 | 6.67 | 8.25 | 5.75 | 1.33 | 40 | 8 | 6.15 |
| MPS | 6 | 2.33 | 7 | 11.5 | 4.33 | 31.17 | 6.23 | 3.44 |
| CC-CMA-ES | 2 | 4.67 | 8 | 4.75 | 9.33 | 28.75 | 5.75 | 2.92 |
| DECC-G | 1 | 2.67 | 3.75 | 6 | 10 | 23.42 | 4.68 | 3.48 |

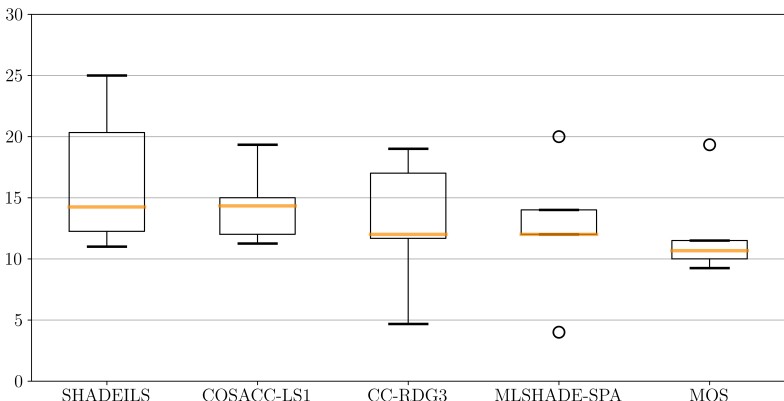

**Figure 4.** Variance of the adjusted scores.

Taking into account the number of problems of each type, we can conclude that the proposed algorithm performs well with all types of LSGO problems. This fact makes COSACC-LS1 preferable in solving "black-box" LSGO problems when information on the problem type is not available. At the same time, COSACC-LS1 proposes a general framework for hybridization of multiple problem decomposition schemes, a global optimizer, and a local search algorithm, thus it has great potential for further improving its performance by applying other component approaches.

## 7. Conclusions

In this paper, a framework for solving LSGO problems has been proposed and a new optimization algorithm COSACC-LS1 based on the framework has been designed and investigated. The performance of COSACC-LS1 has been evaluated and compared with state-of-the-art approaches using the IEEE CEC LSGO 2013 benchmark and the TACO database. The proposed approach outperforms all LSGO competition winners except for one approach—SHADEILS. At the same time, COSACC-LS1 performs well with all types of LSGO problems, while SHADEILS shows poor results on fully-separable problems.

COSACC-LS1 proposed an original hybridization of three main LSGO techniques: CC, DE, and LS. In this work, we have applied SHADE as a DE component, MTS LS1 as LS, and a new approach for the adaptive selection of problem decomposition (several variants with different sizes of subcomponents). The proposed framework does not specify the exact choice of component algorithms, and the user may apply any global and local search algorithm. In that sense, the proposed approach has potential for improvement. In our further research we will examine the proposed framework with other stochastic population-based metaheuristics.

Interaction of three CC-based algorithms demonstrates high performance due to adaptive redistribution of computational resources. We have visualized the redistribution and have found that the approach can adapt to a new environment (new landscape of a LSGO problem). Instead of selecting one variant of decomposition, the interaction allows the component algorithm with the least amount of resources to still participate in the optimization process, and we can see that the algorithm contributes to the optimization process in some regions of the search space.

The well-known "No free lunch" theorem says that it is not possible to choose one optimization algorithm that performs well for all types and instances of optimization problems. At the same time, we can relax the theorem by introducing self-adaptive control of multiple approaches. The approach can adaptively design an effective algorithm (by giving more computations to the best component algorithm) for a specific optimization problem, as well as for a specific region of the search space within the optimization process.

Even though the LSGO benchmark contains many types of LSGO problems, many real-world optimization problems are not well studied and can require fine adjustment of

some COSACC-LS1 parameters. In further work, we will address the issue of developing an approach for online adaptation of the internal parameters of the subcomponent optimizers.

**Author Contributions:** A.V.: methodology, software, validation, investigation, resources, writing—original draft preparation, visualization; E.S. (Evgenii Sopov): conceptualization, methodology, formal analysis, writing—review and editing, visualization, supervision; E.S. (Eugene Semenkin): conceptualization, formal analysis, writing—review and editing, supervision, funding acquisition. All authors have read and agreed to the published version of the manuscript.

**Funding:** This work was supported by the Ministry of Science and Higher Education of the Russian Federation within limits of state contract No. FEFE-2020-0013.

**Institutional Review Board Statement:** Not applicable.

**Informed Consent Statement:** Not applicable.

**Data Availability Statement:** Not applicable.

**Conflicts of Interest:** The authors declare no conflict of interest.

## Nomenclature

The following abbreviations are used in this manuscript:

| | |
|---|---|
| $best\_found_{before}$ | The best found solution before an optimization cycle |
| $best\_found_{after}$ | The best found solution after an optimization cycle |
| $CalculateDiversity(population)$ | Function for calculating the diversity of the population |
| $CC_i$ | The number of subcomponents of the $i$-th algorithm |
| $CC - SHADE(population, NP, i)$ | Function for evolving the population using the cooperative co-evolution algorithm with $NP$ individuals and $i$ subcomponents |
| $Cr$ | Crossover rate |
| $current - to - pbest/1$ | Mutation scheme in SHADE |
| $DI$ | Population diversity |
| $EvalNumGenerations$ | Function for calculating a new number of generations |
| $EvalPopsize$ | Function for calculating a new value of the population size |
| $EvalRD$ | Function for calculating relative diversity of the population |
| $F$ | Scale factor |
| $FEV$ | The number of function evaluations |
| $G_i$ | The number of generations of the $i$-th algorithm |
| $G_{lose}$ | The minimal number of generations |
| $G_{lose}$ | The number of generations by which the budget of algorithms is reduced |
| $G_{win}$ | The number of generations by which the budget of algorithms is increases |
| $GetBestFound$ | Function that returns the best-found solution from the population |
| $GetMedianFitness(population)$ | Function that returns the median fitness value in the population |
| $H$ | The number of $F$ and $Cr$ pairs in SHADE |
| $improvment\_rate_i$ | The change of the best-found fitness of the $i$-th algorithm in an optimization cycle |
| $IR$ | A set of indexes of algorithms with the best improvement rate |
| $M$ | The number of algorithms |
| $maxFEVs$ | The maximum number of fitness function evaluations |
| $maxNP$ | The upper bound for the population size |
| $minNP$ | The lower bound for the population size |
| $medianFitness_{before}$ | The median fitness in the population before an optimization cycle |
| $medianFitness_{after}$ | The median fitness in the population after an optimization cycle |
| $n$ | The number of decision variables |
| $NI$ | The number of algorithms with the best improvement rate |
| $NP$ | The population size |
| $pool$ | The number of generations for redistribution |

| | |
|---|---|
| *RandomPermutation* | Function that randomly permutes values of a vector |
| *RandomPopulation*(*n*, *NP*) | Function that generates a random population with *NP* individuals of *n* variables |
| *RD* | The relative diversity |
| *RFES* | The relative spend of the FEV budget |
| *rRD* | The required value of *RD* |
| *SR*[*i*] | A search range for the *i*-th coordinate in MTS-LS1 |
| *STD* | The standard deviation |
| *u* | A mutant vector |
| $x_{pbest}$ | A random solution chosen from the *p* best individuals |
| $x_t$ | An individual chosen using the tournament selection |

## Acronyms

The following acronyms are used in this manuscript:

| | |
|---|---|
| ABBO | Automated Black-box Optimization |
| BICCA | Bi-space Interactive Cooperative Co-evolutionary Algorithm |
| CABC | Cooperative Artificial Bee Colony |
| CBCC | Contribution Based Cooperative Co-evolution |
| CBFO | Cooperative Bacterial Foraging Optimization |
| CC | Cooperative Co-evolution |
| CC-CMA-ES | Scaling up Covariance Matrix Adaptation Evolution Strategy |
| CCDE | Cooperative Co-evolutionary Differential Evolution |
| CCEA-AVP | Correlation-based Adaptive Variable Partitioning |
| CCFR2 | Extended Cooperative Co-evolution Framework |
| CCGA | Cooperative Co-evolutionary Approach for Genetic Algorithm |
| CC-GDG-CMAES | Competitive Divide-and-conquer Algorithm Covariance Matrix Adaptation Evolution Strategy |
| CCOABC | Cooperative Co-evolution Orthogonal Artificial Bee Colony |
| CCPSO | Cooperatively Co-evolving Particle Swarms Algorithm |
| CC-RDG3 | Cooperative Co-evolution Recursive Differential Grouping |
| CCVIL | Cooperative Co-evolution with Variable Interaction Learning |
| C-DEEPSO | Canonical Differential Evolutionary Particle Swarm Optimization |
| CEC | IEEE Congress on Evolutionary Computation |
| COSACC-LS1 | Coordination of Self-adaptive Cooperative Co-evolution Algorithms with Local Search |
| CPSO | Cooperative Approach to Particle Swarm Optimization |
| CPSO-Hk | Combination of Cooperative Approach to Particle Swarm Optimization with k Subcomponents with the Standard Particle Swarm Optimization |
| CPSO-Sk | Cooperative Approach to Particle Swarm Optimization with k Subcomponents |
| CPU | Central Processing Unit |
| DE | Differential Evolution |
| DECC-DG | Cooperative Co-Evolution with Differential Grouping |
| DECC-DG2 | Cooperative Co-Evolution with A Faster and More Accurate Differential Grouping |
| DECC-DML | Cooperative Co-evolution with Delta Grouping |
| DECC-G | Self-Adaptive Differential Evolution with Neighborhood Search with Cooperative Co-evolution |
| DECC-ML | Multilevel Cooperative Co-evolution with More Frequent Random Grouping |
| DECC-XDG | Cooperative Co-Evolution with Extended Differential Grouping |
| DGSC | Differential Grouping with Spectral Clustering |
| DIMA | Dependency Identification with Memetic Algorithm |
| DMS-PSO | Dynamic Multi-Swarm Particle Swarm Optimizer |
| EA | Evolutionary Algorithm |
| FEPCC | Fast Evolutionary Programming with Cooperative Co-evolution |
| GA | Genetic Algorithm |

| IHDELS | Iterative Hybridization of Differential Evolution with Local Search |
| IPOPCMA-ES | Restart Covariance Matrix Adaptation Evolution Strategy with Increasing Population Size |
| IRRG | Incremental Recursive Ranking Grouping |
| JADE | Adaptive Differential Evolution with Optional External Archive |
| L-BFGSB | Limited-memory the Broyden–Fletcher–Goldfarb–Shanno algorithm |
| LSGO | Large-Scale Global Optimization |
| L-SHADE | Iteratively Applies Success History-Based Differential Evolution with Linear Population Size Reduction |
| MLCC | Multilevel Cooperative Co-evolution |
| MLSHADE-SPA | Memetic Framework for Solving Large-scale Optimization Problems |
| MOS | Multiple Offspring Sampling |
| MPICH2 | Message Passing Interface Chameleon |
| MPS-CMA-ES | Hybrid of Minimum Population Search and Covariance Matrix Adaptation Evolution Strategy |
| MTS | Multi-trajectory Search |
| MWW | Mann-Whitney-Wilcoxon |
| NSGA-2 | Non-dominated Sorting Genetic Algorithm |
| PSO | Particle Swarm Optimization |
| SACC | Cooperative Co-evolution with Sensitivity Analysis-based Budget Assignment Strategy |
| SaDE | Self-Adaptive Differential Evolution |
| SaNSDE | Self-Adaptive Differential Evolution with Neighborhood Search |
| SGCC | Cooperative Co-evolution with Soft Grouping |
| SHADE-ILS | Success-History Based Parameter Adaptation for Differential Evolution with Iterative Local Search |
| TACO | Toolkit for Automatic Comparison of Optimizers |
| VMODE | Variable Mesh Optimization Differential Evolution |

## Appendix A. Plots of Convergence, the Population Size, and Redistribution of the Computational Resources

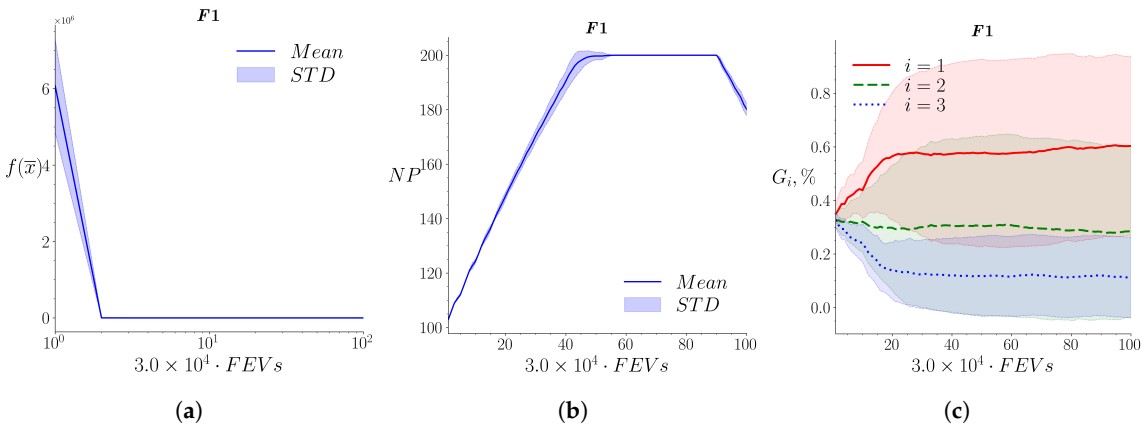

**Figure A1.** The dynamics of COSACC-LS1 on the F1 problem: (**a**) Convergence; (**b**) Population size; (**c**) Redistribution of resources.

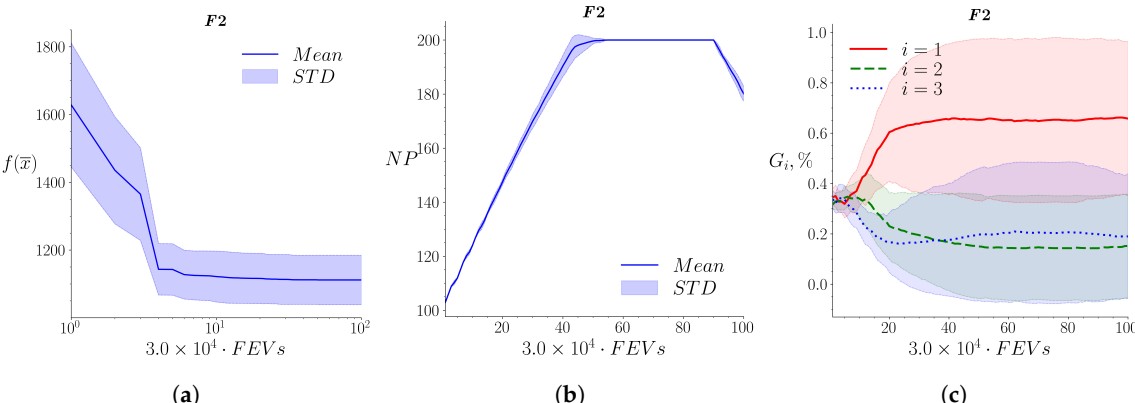

(**a**)  (**b**)  (**c**)

**Figure A2.** The dynamics of COSACC-LS1 on the F2 problem: (**a**) Convergence; (**b**) Population size; (**c**) Redistribution of resources.

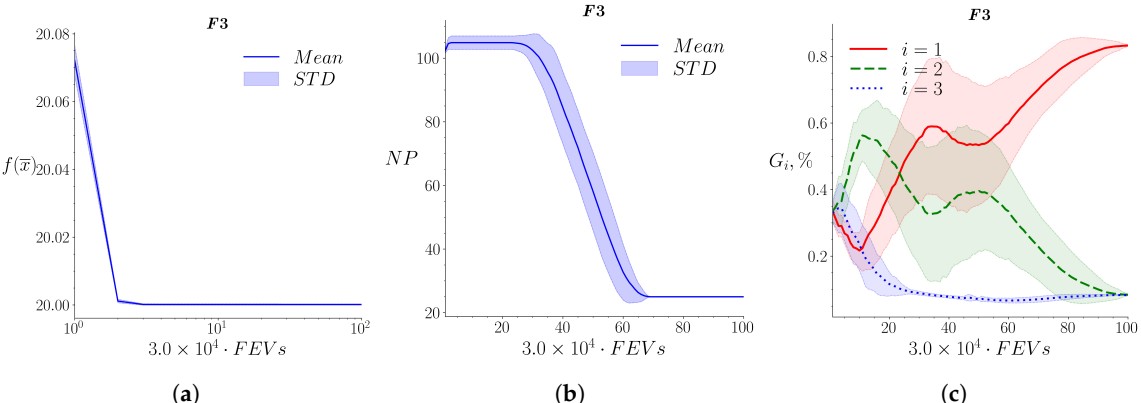

(**a**)  (**b**)  (**c**)

**Figure A3.** The dynamics of COSACC-LS1 on the F3 problem: (**a**) Convergence; (**b**) Population size; (**c**) Redistribution of resources.

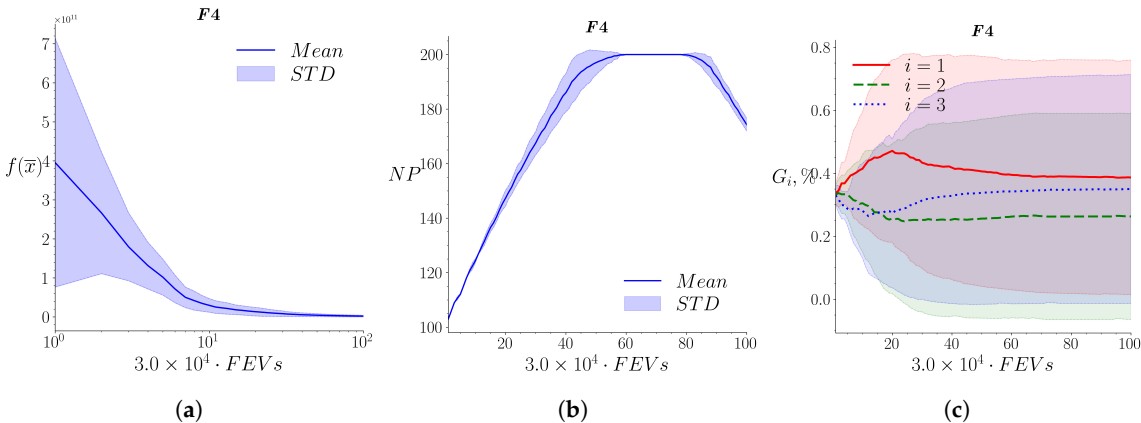

(**a**)  (**b**)  (**c**)

**Figure A4.** The dynamics of COSACC-LS1 on the F4 problem: (**a**) Convergence; (**b**) Population size; (**c**) Redistribution of resources.

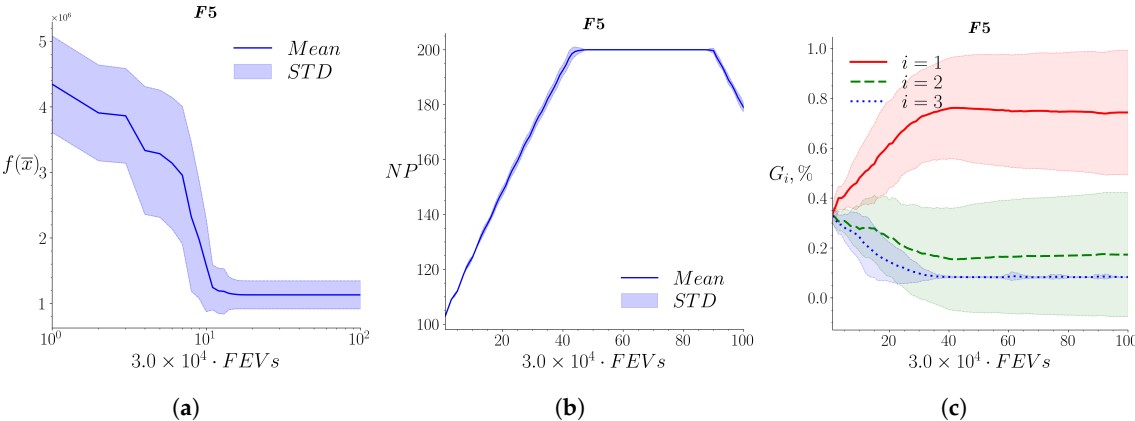

**Figure A5.** The dynamics of COSACC-LS1 on the F5 problem: (**a**) Convergence; (**b**) Population size; (**c**) Redistribution of resources.

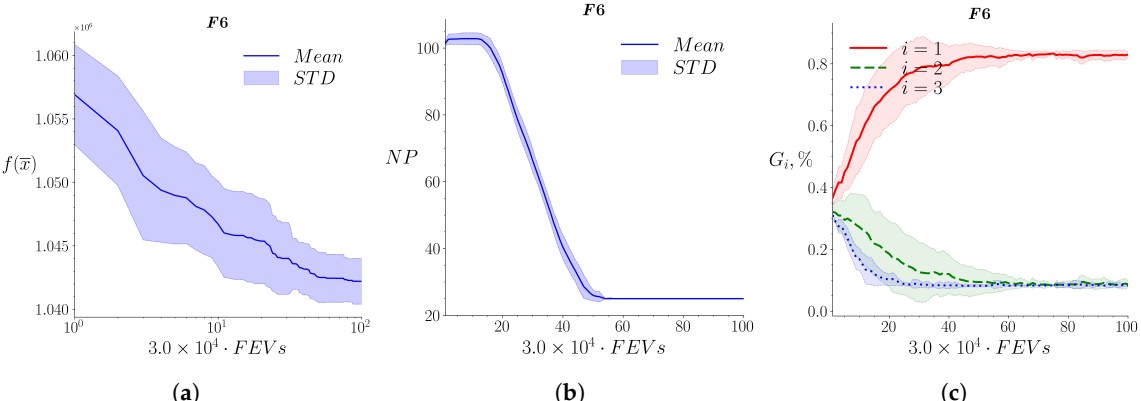

**Figure A6.** The dynamics of COSACC-LS1 on the F6 problem: (**a**) Convergence; (**b**) Population size; (**c**) Redistribution of resources.

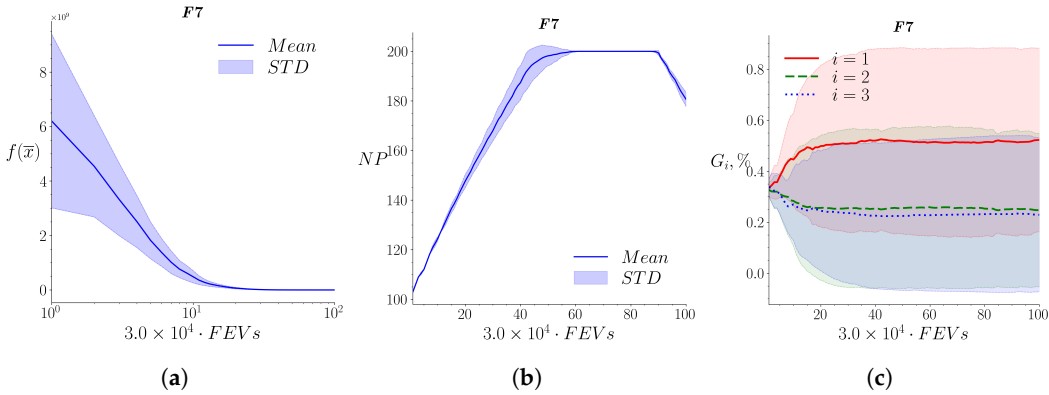

**Figure A7.** The dynamics of COSACC-LS1 on the F7 problem: (**a**) Convergence; (**b**) Population size; (**c**) Redistribution of resources.

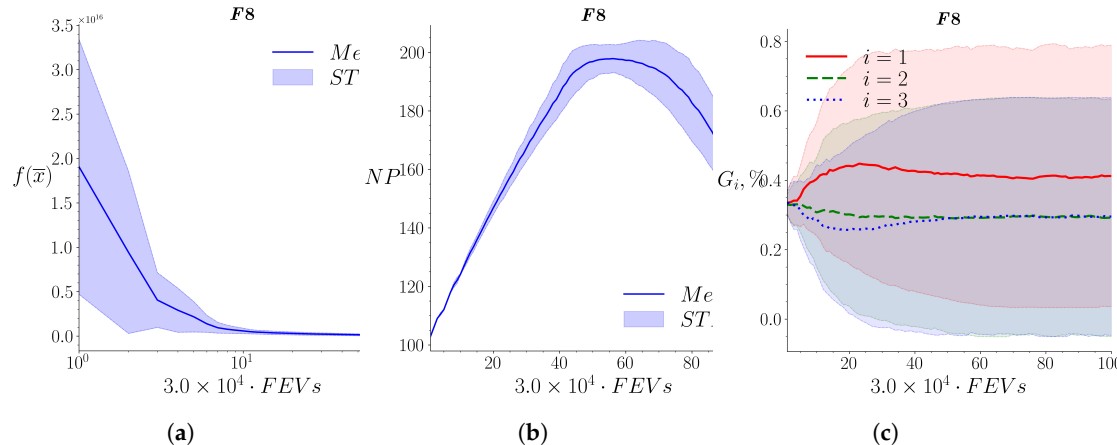

**Figure A8.** The dynamics of COSACC-LS1 on the F8 problem: (**a**) Convergence; (**b**) Population size; (**c**) Redistribution of resources.

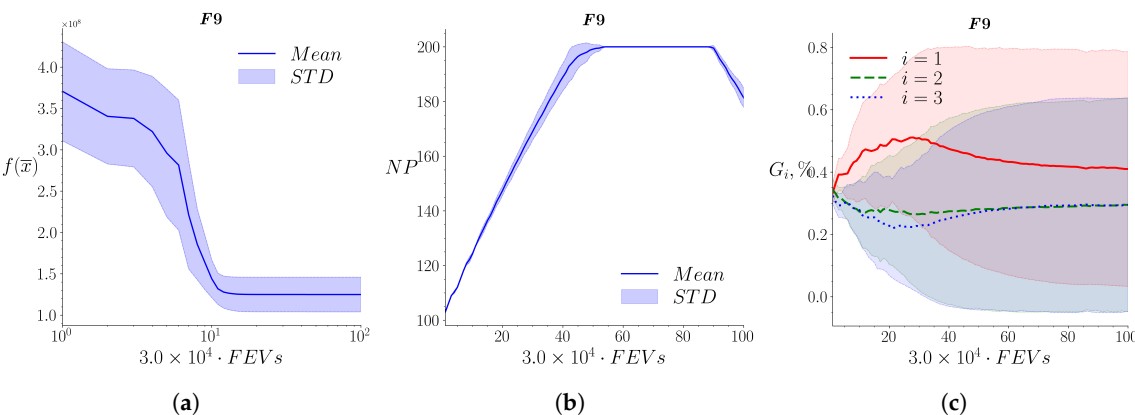

**Figure A9.** The dynamics of COSACC-LS1 on the F9 problem: (**a**) Convergence; (**b**) Population size; (**c**) Redistribution of resources.

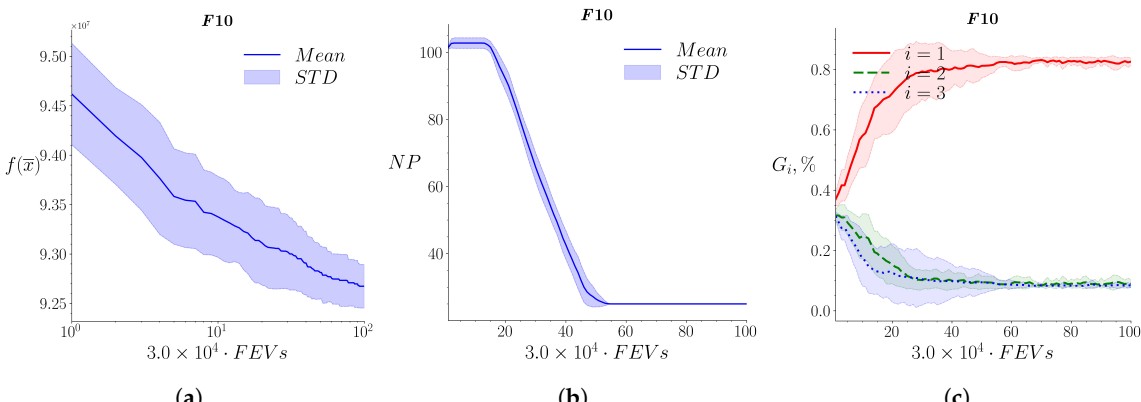

**Figure A10.** The dynamics of COSACC-LS1 on the F10 problem: (**a**) Convergence; (**b**) Population size; (**c**) Redistribution of resources.

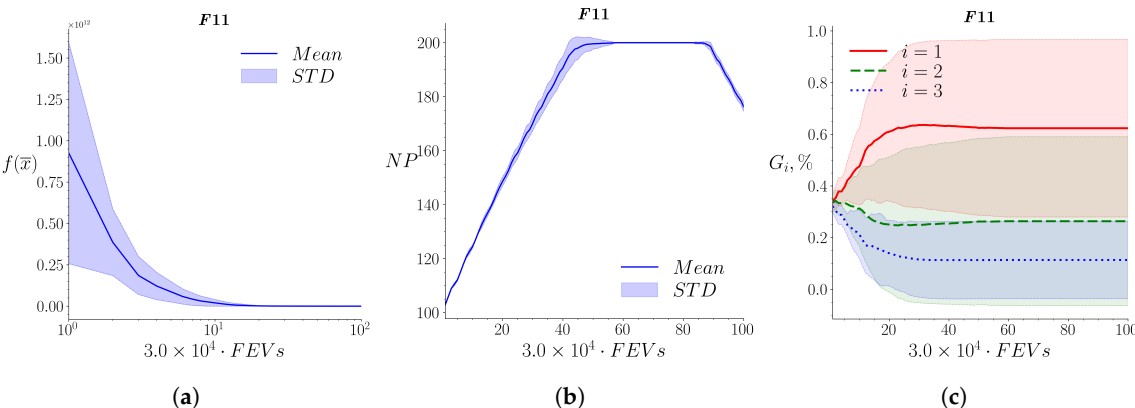

**Figure A11.** The dynamics of COSACC-LS1 on the F11 problem: (**a**) Convergence; (**b**) Population size; (**c**) Redistribution of resources.

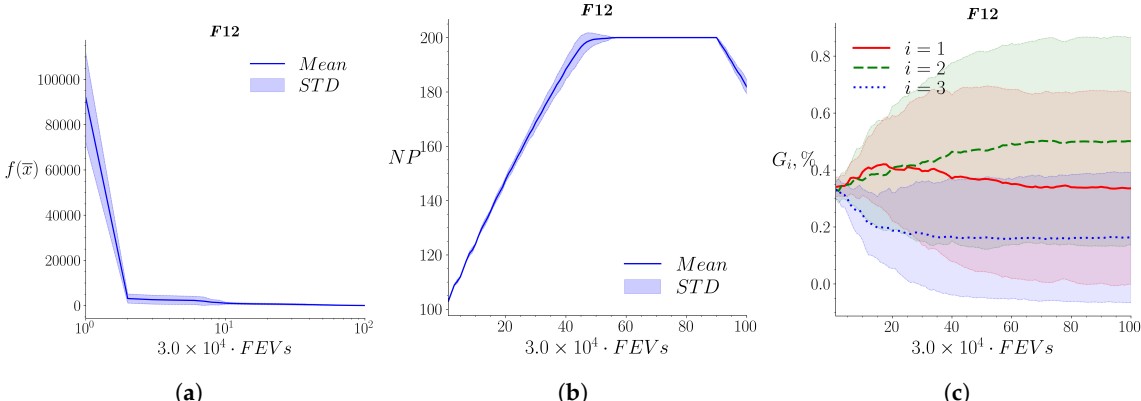

**Figure A12.** The dynamics of COSACC-LS1 on the F12 problem: (**a**) Convergence; (**b**) Population size; (**c**) Redistribution of resources.

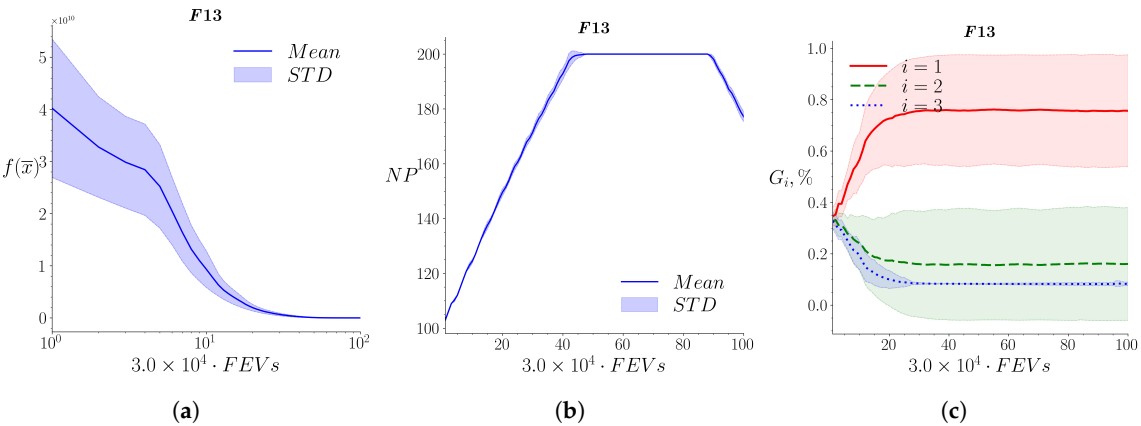

**Figure A13.** The dynamics of COSACC-LS1 on the F13 problem: (**a**) Convergence; (**b**) Population size; (**c**) Redistribution of resources.

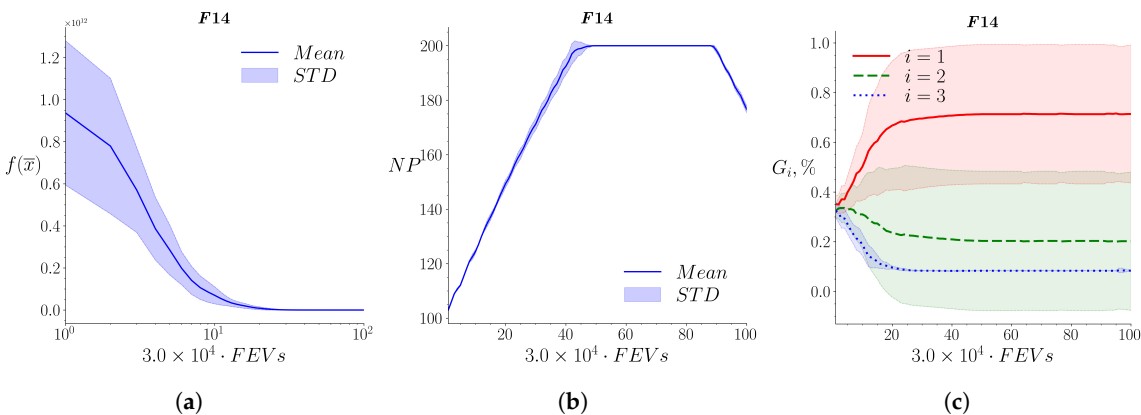

**Figure A14.** The dynamics of COSACC-LS1 on the F14 problem: (**a**) Convergence; (**b**) Population size; (**c**) Redistribution of resources.

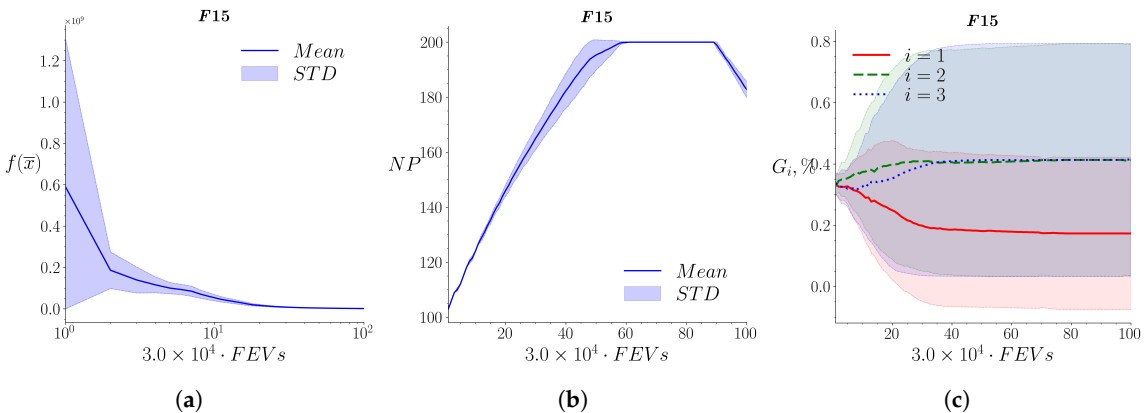

**Figure A15.** The dynamics of COSACC-LS1 on the F15 problem: (**a**) Convergence; (**b**) Population size; (**c**) Redistribution of resources.

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
