# Peer review of "On Improving Adaptive Problem Decomposition Using Differential Evolution for Large-Scale Optimization Problems"

_mathematics, doi:10.3390/math10224297_

Round 1

Reviewer 1 Report

Dear Authors:

Thank you for your excellent paper. It is very well prepared and written.

The Introduction and literature review is convencing and suffienct.

The proposed approach combines multiple ideas from state-of-the-art algorithms and implements "Coordination of Self-adaptive Cooperative Coevolution algorithms with Local Search". The developed self-adaptation method provides tuning of both the structure of the complete approach and the parameters of each algorithm in the cooperation.

The performance of COSACC-LS1 has been investigated using the CEC LSGO 2013 benchmark and the results have been been compared with the results of some known leading LSGO approaches. It has been clearly shown in the results that the application of the proposed approach provides better performance than all LSGO competition winners except for SHADEILS. 

However, consider the following two simple comments:

1- Figures 3,4, and 5 are repeated in the appendix, there is no need to repeat them.

2- Avoid the extensive use of "we..." in describing the application of your procedure or work.  For instance, you can use passive voice. 

Best wishes

Author Response

We would like to thank the reviewer for the high appreciation of our work and for constructive and useful comments.

We have revisited our paper and have made the following changes:

Remark 1. Figures 3,4, and 5 are repeated in the appendix, there is no need to repeat them.

We have removed these figures from the main section and have made references to figures from the appendix.

Remark 2. Avoid the extensive use of "we..." in describing the application of your procedure or work.  For instance, you can use passive voice.

We have made the corresponding changes in the Abstract section and in the majority of parts of the main section, where it doesn’t affect the meaning, we want to convey.

Reviewer 2 Report

The article presented a novel approach for improving the adaptive problem decomposition using a method based on differential evolution for large-scale optimization problems. The manuscript is well written, structured, and organized. I recommend the acceptance of the manuscript. For the final version, please consider the following minor observations:

1. Please revise again all mathematical formulas. A few cases which require special attention are: (page 7, line 270) and (page 9, line 350) where it is not clear if the elements are all different or if only pairs of elements are different.

2. As a possible future research direction, please check how the proposed method can be adapted for algorithms similar with differential evolution. Please include those observations in the Discussion section or in the Conclusions section.

3. Please improve the quality of the Figure 3, Figure 4, Figure 5, and Figure 6 for readability such that the font size of the text is higher.

Author Response

We would like to thank the reviewer for the high appreciation of our work and for constructive and useful comments.

We have revisited our paper and have made the following changes:

Remark 1. Please revise again all mathematical formulas. A few cases which require special attention are: (page 7, line 270) and (page 9, line 350) where it is not clear if the elements are all different or if only pairs of elements are different.

We have added information on that all elements are to be different.

Remark 2. As a possible future research direction, please check how the proposed method can be adapted for algorithms similar with differential evolution. Please include those observations in the Discussion section or in the Conclusions section.

In the Conclusion section, we have previously mentioned that “The proposed framework does not specify the exact choice of component algorithms, and the user may apply any global and local search algorithm.” We agree with your recommendation, and we have pointed out that in our further research we will examine the framework with other metaheuristics.

Remark 3. Please improve the quality of the Figure 3, Figure 4, Figure 5, and Figure 6 for readability such that the font size of the text is higher.

We have renewed all figures using the increase font size.

Reviewer 3 Report

The authors propose a novel approach to deal with large-scale optimization problems. The manuscript is well-written and organized. The authors made a substantial literature review and present their results in a rigorous way. Nonetheless, there are some details that the authors might want to consider for improving the presentation of the document.

1-    Please clearly state in the abstract the main contribution of your research

2-    Please expand a little more Section I (introduction) to give a more in-depth explanation of the problem at hand and the main challenges faced by the authors in this research.

3-    Section II (related work) presents a fairly good recompilation and analysis of the specialized literature. Please complement this section with a table that contrasts the main features of previous research work.

4-    Please evaluate the possibility of including the following papers in Section II  

10.1109/TEVC.2021.3108185

10.1109/TEVC.2022.3216968

5-    A list of acronyms is necessary at the end of the document

6-    Please enhance the quality of figure 6

7-    Please change line 12 in the introduction: instead of “we have proposed” use “we propose”

8-    Please include a nomenclature at the end of the document.

Author Response

We would like to thank the reviewer for the high appreciation of our work and for constructive and useful comments.

We have revisited our paper and have made the following changes:

Remark 1. Please clearly state in the abstract the main contribution of your research

We have added the following text to stress the contribution of our work “The main contribution of the study is a new self-adaptive approach that is preferable for solving hard real-world problems because it is not overfitted with the LSGO benchmark due to self-adaptation during the search process instead of a manual benchmark-specific fine-tuning.”

Remark 2. Please expand a little more Section I (introduction) to give a more in-depth explanation of the problem at hand and the main challenges faced by the authors in this research.

We have added the following text in the Introduction section. “As previously mentioned, the performance of many black-box global optimization algorithms can’t be improved by only increasing the budget of function evaluations when solving LSGO problems. One of challenges for researchers in the LSGO field is the development of new approaches, which can deal with the high dimensionality. Various LSGO algorithms that use fundamentally different ideas and demonstrate different performances for different classes of LSGO problems have been proposed. When solving a specific LSGO problem, a researcher must choose an appropriate LSGO algorithm and fine-tune its parameters. Moreover, the algorithm can require different settings at different states of the optimization process (for example, at exploration and exploitation stages). Thus, the development of self-adaptive approaches for solving hard LSGO problem is an actual research task.”

Remark 3.    Section II (related work) presents a fairly good recompilation and analysis of the specialized literature. Please complement this section with a table that contrasts the main features of previous research work.

We have summarized all reviewed LSGO approaches in a table and have highlighted their main features, namely, decomposition types, global and local search algorithms used.

Remark 4.    Please evaluate the possibility of including the following papers in Section II 

Thank you for this recommendation. These papers have been included in the review section. We will investigate in depth the results from these papers and use them in our further research.

Remark 5. A list of acronyms is necessary at the end of the document

We have added the Abbreviations section (using the MDPI template).

Remark 6. Please enhance the quality of figure 6

We have renewed all figures; font sizes are increased.

Remark 7. Please change line 12 in the introduction: instead of “we have proposed” use “we propose”

The change is made.

Remark 8. Please include a nomenclature at the end of the document.

We have added the Nomenclature section.